# Abstraction between Structural Causal Models:
# A Review of Definitions and Properties

**Fabio Massimo Zennaro**[1]

[1]University of Warwick, Coventry, United Kingdom

## Abstract

Structural causal models (SCMs) are a widespread formalism to deal with causal systems. A recent direction of research has considered the problem of relating formally SCMs at different levels of abstraction, by defining maps between SCMs and imposing a requirement of interventional consistency. This paper offers a review of the solutions proposed so far, focusing on the *formal properties* of a map between SCMs, and highlighting the different *layers* (structural, distributional) at which these properties may be enforced. This allows us to distinguish families of abstractions that may or may not be permitted by choosing to guarantee certain properties instead of others. Such an understanding not only allows to distinguish among proposal for causal abstraction with more awareness, but it also allows to tailor the definition of abstraction with respect to the forms of abstraction relevant to specific applications.

## 1 INTRODUCTION

Modelling causal systems at different levels of abstraction is a central feature in our understanding of the world and in our scientific endeavours. Generating representations at the right level of granularity is often the implicit goal of unsupervised and representation learning; in the causal setup, learning high-level causal representations is the central task of causal representation learning (CRL) [Schölkopf et al., 2021].

A study of the formal properties of abstraction in the context of structural causal models (SCM) has been proposed in a few recent papers [Rubenstein et al., 2017, Beckers and Halpern, 2019, Beckers et al., 2020, Rischel, 2020, Rischel and Weichwald, 2021, Otsuka and Saigo, 2022]. All these works have in common the presentation of abstraction as a map between SCMs that would be consistent (or approxi-

mately consistent) with respect to interventions; that is, a map that would commute (or approximately commute) with respect to interventions, thus enforcing the intuition that working first at the microlevel and then abstracting to the macrolevel would produce the same result as immediately switching to the macrolevel and then working there.

Yet SCMs are complex structured objects and the abstraction maps proposed in the existing literature vary considerably between them. This is reflected in the different nomenclature (abstraction, transformation), theoretical background (measure theory, category theory) and concrete mappings that are allowed to be considered as abstractions. In this paper, we focus only on definitions of abstractions, without discussing explicitly the requirement of (interventional) consistency. The objective is to reconcile existing formalisms in terms of the *layers* at which an abstraction is defined and the *properties* that are enforced. This will bring some immediate benefits, among which: (i) a better understanding of the alternative formalisms being proposed; (ii) an insight into which maps we would want to be considered as abstractions; (iii) in a CRL framework, the ability to tailor the definition according to the properties we actually want to hold.

The paper is organized as follows: in Section 2 we review the definition of SCM and abstraction, and we summarize relevant work in the literature; in Section 3 we analyze how abstractions can be characterized on two different layers, and which properties can be required; finally, in Section 4, we discuss the implications of our analysis, summarizing how properties on different layers may be used to restrict the definition of an abstraction.

## 2 BACKGROUND

### 2.1 STRUCTURAL CAUSAL MODELS

A (semi-Markovian) SCM Pearl [2009], Peters et al. [2017] $\mathcal{M}$ is a tuple $\langle \mathcal{X}, \mathcal{U}, \mathcal{F}, P(\mathcal{U}) \rangle$ where:

*Accepted for the Causal Representation Learning workshop at the 38th Conference on Uncertainty in Artificial Intelligence* (UAI CRL 2022).

- $\mathcal{X}$ is a set of $N$ endogenous variables, that is, observed variables of interest in the model; each variable $X_i \in \mathcal{X}$ is defined on a finite domain $\mathcal{M}[X_i]$;

- $\mathcal{U}$ is a set of $N$ exogenous variables, that is, unobserved variables accounting for factors beyond the scope of the model; each variable $U_i \in \mathcal{U}$ is defined on a domain $\mathcal{M}[U_i]$;

- $\mathcal{F}$ is a set of $N$ modular and measurable structural functions, one for each endogenous variable $X_i$; the value assumed by an endogenous node is deterministically computed as $X_i = f_i(pa(X_i), U_i)$, where $pa(X_i) \subseteq \mathcal{X} \setminus \{X_i\}$ is the set of endogenous variables directly affecting $X_i$;

- $P(\mathcal{U})$ is a probability distributions over the exogenous variables.

A SCM admits an underlying acyclic graph $\mathcal{G}_{\mathcal{M}} = \langle V, S \rangle$ whose set of vertices $V$ is given by the endogenous variables in $\mathcal{X}$ and the set of edges $E$ is given by the structural functions [Pearl, 2009].

Our definition of SCM makes a few assumption about the causal model such as: (a) finite number of endogenous variables; (b) finite domains for the endogenous variables; (c) unique exogenous variable per endogenous variable; (d) exogenous variables not necessarily independent; (e) modularity and measurability of structural functions; (f) acyclic underlying graph. All these assumptions are discussed at length in the literature [Pearl, 2009, Peters et al., 2017], and they represent a minimum shared set of assumptions between the existing proposals for abstraction. In the following, we will therefore assume that our SCMs always comply with these assumptions, unless otherwise stated.

Thanks to the assumptions above, it is possible to pushforward probability distributions from the exogenous variables onto the endogenous variables. Furthermore, we can conveniently represent a SCM as a collection of sets and a collection of mechanisms $\mathcal{M}[\phi_{X_i}]$, one for each endogenous variable, encoded as a stochastic matrix (or Markov kernel) [Rischel, 2020].

SCMs can be modified via intervention through the $\iota : do(\mathbf{X} = \mathbf{x})$ operator which fixes a set of endogenous variables $\mathbf{X} \subseteq \mathcal{X}$ to the values $\mathbf{x}$ [Pearl, 2009]. An intervention $\iota$ on a SCM $\mathcal{M}$ produces a new post-interventional model $\mathcal{M}_\iota$.

## 2.2 ABSTRACTIONS

Given two SCMs $\mathcal{M}^m = \langle \mathcal{X}, \mathcal{U}, \mathcal{F}, P(\mathcal{U}) \rangle$ and $\mathcal{M}^M = \langle \mathcal{X}', \mathcal{U}', \mathcal{F}', P(\mathcal{U}') \rangle$, an abstraction is a map $\alpha : \mathcal{M}^m \to \mathcal{M}^M$ between the two SCMs. For simplicity, we will refer to the model $\mathcal{M}^m$ in the domain of $\alpha$ as the micromodel, or the low-level model, and to the model $\mathcal{M}^M$ in the codomain of $\alpha$ as the macromodel, or the high-level model.

Characterizing an abstraction requires a concrete specification of the map $\alpha : \mathcal{M}^m \to \mathcal{M}^M$, together with the properties that this map must satisfy. We can distinguish two forms of properties that are usually required: (i) formal properties regarding the map itself, that statically constrain the map (e.g.: surjectivity); and (ii) consistency properties guaranteeing that working either with the micromodel or the macromodel would lead to consistent results. The requirement of consistency often takes the form of (perfect or approximate) commutativity with respect to interventions, which translates the understanding that the results observed from the micromodel and the macromodel should be consistent when we intervene on them. In the next sections we will focus on formal properties, leaving a detailed study of the property of consistency for future work.

## 2.3 RELATED WORK

A first definition of abstraction between SCMs (for which assumption (b) and (f) are not necessarily required) has been put forth by Rubenstein et al. [2017]: a $(\tau\text{-}\omega)$-*transformation* is a map $\tau : \mathcal{M}^m[\mathcal{X}] \to \mathcal{M}^M[\mathcal{X}']$ from the joint domain of the outcomes of all the microvariables to the joint domain of the outcomes of all the macrovariables, which is interventionally consistent with respect to a set of interventions of interest $\mathcal{I}$.

Dealing with deterministic SCMs, Beckers and Halpern [2019] build over this work by presenting the notion of *uniform $(\tau\text{-}\omega)$-transformation* as a map $\tau : \mathcal{M}^m[\mathcal{X}] \to \mathcal{M}^M[\mathcal{X}']$ that would satisfy interventional consistency for any choice of distribution over the exogenous variables in $\mathcal{M}^m$. Further, they also offer a stronger defintion of $\tau$-*abstraction* as a surjective function $\tau : \mathcal{M}^m[\mathcal{X}] \to \mathcal{M}^M[\mathcal{X}']$ with an associated surjective function between exogenous nodes of the micromodel and the macromodel, and an associated function between set of interventions in the micromodel and the macromodel.

Relying on category theory, Rischel [2020], Rischel and Weichwald [2021] suggest a more detailed definition of an abstraction. An abstraction $\alpha$ is a a tuple $\langle R, a, \alpha_{X'} \rangle$ given by a set of relevant nodes in the micromodel $R \subseteq \mathcal{X}$, a surjective map between variables in the micromodel and macromodel $a : R \to \mathcal{X}'$, and a collection of surjective maps $\alpha_{X'}$, one for each macrovariable: $\alpha_{X'} : \mathcal{M}^m[a^{-1}(X_i')] \to \mathcal{M}^M[X_i']$; consistency is evaluated with respect to interventions.

Finally, Otsuka and Saigo [2022] offers an alternative category-theoretical definition, by requiring an abstraction $\alpha$ to be defined by first finding a graph homomorphism from $\mathcal{G}_{\mathcal{M}^m}$ to $\mathcal{G}_{\mathcal{M}^M}$, expressing the micromodel $\mathcal{M}^m$ and the macromodel $\mathcal{M}^M$ as functors, and finally by seeing the abstraction as a natural transformation between the two functors. Interventions, once again used to enforce interventional

consistency, take the form of endofuctors.

# 3 CHARACTERIZING ABSTRACTIONS

In the previous section we have briefly reviewed different specifications of abstractions. Here, we want to bring all these definitions together by reviewing how they satisfy different types of properties on different layers.

## 3.1 LAYERS OF AN ABSTRACTION

A SCM is a mathematical object that brings together two types of information: the statistical data-driven behaviour of a set of variables in a given model (which could be either the pre-interventional $\mathcal{M}$ or a post-interventional $\mathcal{M}_\iota$), and the causal assumption-driven structure connecting variables (through causal links) and models (through interventions).

Although these two types of information are not independent, it can be useful, when specifying an abstraction, to define explicitly how the abstraction behaves with respect to them. We can therefore distinguish these two layers:

1. A *structural* layer which deals with a map $\mathcal{G}_{\mathcal{M}^m} \to \mathcal{G}_{\mathcal{M}^M}$, that is, how the underlying graph $\mathcal{G}_{\mathcal{M}^m}$ of a micromodel is transformed through the abstraction $\boldsymbol{\alpha}$. This layer is concerned with maps between nodes and, possibly, edges. The structural layer accounts for how an abstraction can transfrom the identity of the causes, and how it can affect flow and the directionality of causes and effects.

2. A *distributional* layer which deals with maps $\mathcal{M}^m[\mathbf{X}] \to \mathcal{M}^M[\mathbf{X}'], \mathbf{X} \subseteq \mathcal{X}, \mathbf{X}' \subseteq \mathcal{X}'$, that is, how outcomes and distributions implied by a micromodel are transformed through the abstraction $\boldsymbol{\alpha}$. This layer may define a single map $\mathcal{M}^m[\mathcal{X}] \to \mathcal{M}^M[\mathcal{X}']$ relating the whole joint outcome spaces of the models [Rubenstein et al., 2017, Beckers and Halpern, 2019], or a collection of maps, such as $\mathcal{M}^m[X_i] \to \mathcal{M}^M[X_i']$, relating the outcome space of single or subsets of variables [Rischel, 2020]. These maps implicitly define pushforward maps from the probability measure over the set $\mathcal{M}^m[\mathbf{X}]$ onto the probability measure over the set $\mathcal{M}^M[\mathbf{X}']$. The distributional layer thus accounts for how an abstraction can affect the representation and the strengths of relationships of cause and effect.

This separation of concerns is already implicitly present in Rischel [2020], Rischel and Weichwald [2021] where an abstraction is defined on two layers, as a mapping $a$ between variables and a collection of mappings $\alpha_{X'}$ between outcomes. This distinction is given stronger emphasis in Otsuka and Saigo [2022] where an explicit mapping between graphs (via a graph homomorphism) and a mapping between outcomes (via a natural transformation) are required; this setup

follows from the category-theoretical approach of representing a model (e.g., a casual model in Jacobs et al. [2019] or a database in Spivak [2014]) at a syntactic level capturing the underlying structure (in a free category generated from a graph) and at a semantic level (as a functor to a category that instantiates specific values).

## 3.2 PROPERTIES OF AN ABSTRACTION

Relying on the distinction between the two layers above, we now analyze what specific properties may be required on each layer, and what forms of abstraction they entail.

**Properties on the structural layer.** Let us first consider a map $\mathcal{G}_{\mathcal{M}^m} \to \mathcal{G}_{\mathcal{M}^M}$. First of all, notice that between two graphs we can typically establish a *function on nodes* $\mathcal{X}^m \to \mathcal{X}^M$, or a stricter *structure-preserving functor* $\mathcal{C}_{\mathcal{M}^m} \to \mathcal{C}_{\mathcal{M}^M}$, where $\mathcal{C}_{\mathcal{M}^m}, \mathcal{C}_{\mathcal{M}^M}$ are the free categories generated from the DAGs of the micromodel and the macromodel [Otsuka and Saigo, 2022]. Other solutions, such as a *function on edges*, are not considered in the literature. Let us now analyze these two approaches and the connected properties, starting from the functional map:

- *Functionality:* a function on the nodes requires a mapping of all the nodes of $\mathcal{G}_{\mathcal{M}^m}$ onto the nodes of $\mathcal{G}_{\mathcal{M}^M}$. This is a property expressing the requirement that each node in the micromodel is abstracted, and no node can be ignored. This translates the idea of abstraction as a strict coarsening of the variables in a micromodel; structural functionality is implied by functoriality in Otsuka and Saigo [2022] .
  On the other hand, dropping this requirement means that microvariables may just be ignored; this may happen when we want to consider abstractions that capture or synthesize only a sub-part of a micromodel, discarding irrelevant information or indirectly relegating it over the exogenous variables; this is the case in Rischel [2020] where the mapping $a$ between variables has, as its domain, the restriction of $\mathcal{X}^m$ to relevant variables $R \subseteq \mathcal{X}^m$.

- *Functional surjectivity:* functional surjectivity requires a mapping such that all the nodes of $\mathcal{G}_{\mathcal{M}^M}$ are mapped from the nodes of $\mathcal{G}_{\mathcal{M}^m}$. This reflects the understanding that all the variables in a macromodel are explained by one or more variables in the micromodel. Functional surjectivity is explicitly required by Rischel [2020].
  Dropping this condition is equivalent to accepting that a macromodel may include variables that have no explanation in the micromodel; this allows for forms of abstractions akin to compression and embedding, or cases in which exogenous factors of variance in the micromodel have implicitly become endogenous in the macromodel; this choice is made in Rubenstein et al. [2017] and Otsuka and Saigo [2022].

- *Functional injectivity:* functional injectivity requires a mapping such that all the nodes of $\mathcal{M}^m$ are mapped to distinct nodes of $\mathcal{M}^M$. This would encode the desideratum that our abstraction guarantees that no variables are collapsed together. Functional injectivity allows for forms of abstractions such as embeddings, or abstractions in which variables are identical, but their domains and dynamics may be simplified. No work in the literature enforces this property. Without functional injectivity, coarsening and collapsing of variables from the micromodel to the macromodel is allowed.

- *Functional bijectivity:* a trivial property of functional bijection follows by enforcing surjectivity and injectivity. This would allow only for abstractions where there is an isomorphism (an identity or a permutation) between the nodes. No work in the literature enforces this property.

A functorial map implies a more structured map, and it may allow for analogous properties:

- *Functoriality:* a structure-preserving functor requires an explicit mapping of nodes and edges in $\mathcal{M}^m$, such that composition is preserved[1]. This is a strong requirement implying that we want to deal only with abstractions that preserve the directionality of causes and that do not arbitrarily drop any causal link; this is enforced in Otsuka and Saigo [2022] by requiring a graph homomorphism between $\mathcal{G}_{\mathcal{M}^m}$ and $\mathcal{G}_{\mathcal{M}^M}$.
  Violating this property means that we accept abstractions that may arbitrarily drop some causal connections or even reverse them; this may be acceptable for some types of abstractions, such as macromodels that may ignore causal connections with a strength under a certain threshold [Janzing et al., 2013]. Non-functoriality is accepted in Rischel [2020], where the mapping $a$ between variables does not express any constraints between edges in the micromodel and macromodel.

- *Functorial fullness:* functorial fullness requires surjectivity between the sets of edges of the micromodel and the macromodel; that is, if $X_i'$ and $X_j'$ are two nodes in the macromodel mapped respectively from $X_i$ and $X_j$, then we want a surjective map between edges going from $X_i$ to $X_j$ in the micromodel and edges going from $X_i'$ to $X_j'$ in the macromodel. This reflects the understanding that every causal link, either direct or mediated (given by composition of edges), in the macromodel must have a corresponding causal link in the micromodel. No work in the literature enforces this property. No fullness implies the possible presence of additional relationships of cause and effects in the macromodel which are absent in the micromodel.

- *Funtorial faithfulness:* functorial faithfullness requires injectivity between the sets of edges of the micromodel and the macromodel; that is, if $X_i'$ and $X_j'$ are two nodes in the macromodel mapped respectively from $X_i$ and $X_j$, then we want an injective map between edges going from $X_i$ to $X_j$ in the micromodel and edges going from $X_i'$ to $X_j'$ in the macromodel. This implies that every causal link, either direct or mediated (given by composition of edges), in the micromodel has a corresponding causal link in the macromodel. No work in the literature enforces this property. Without functorial faithfulness, collapsing of edges is possible.

- *Functorial fully faithfulness:* a trivial property of fully faithfulness follows by enforcing fullness and faithfulness. Together with bijectivity on nodes, this would allow only for strictly structure-preserving abstractions expressing an isomorphism (an identity) between nodes and edges. No work in the literature enforces this property.

A derived property that is sometimes discussed is the *invertibility* of the map $\mathcal{G}_{\mathcal{M}^m} \to \mathcal{G}_{\mathcal{M}^M}$; it immediately follows that functional bijectivity allows perfect invertibility on nodes; functional surjectivity allows set-invertibility on nodes; functorial fully faithfulness allows perfect invertibility on edges; and, functorial fullness allows set-invertibility on edges[2].

Finally, there are two important properties that are commonly taken for granted, but which could be changed or relaxed, thus providing abstractions with different degrees of freedom:

- *Micro-to-macro:* it seems a reasonable assumption that the directionality of an abstraction should be $\mathcal{G}_{\mathcal{M}^m} \to \mathcal{G}_{\mathcal{M}^M}$, as we normally move from micromodels to macromodels by reducing their complexity. In some contexts, it may be of interest to consider possibly stochastic maps $\mathcal{G}_{\mathcal{M}^M} \to \mathcal{G}_{\mathcal{M}^m}$ going from a macromodel to a micromodel.

- *Determinism:* the mapping from the microlevel to the macrolevel is often interpreted as a deterministic supervenient map. However, in case of limited knowledge and uncertainty, the overall requirement of functoriality or functionality may be dropped by requiring the map $\mathcal{G}_{\mathcal{M}^m} \to \mathcal{G}_{\mathcal{M}^M}$ to be a stochastic map. This would allow for forms of abstraction in which the contribution of a microvariable may be split among a collection of macrovariables. This actually happens in Rubenstein et al. [2017], where outcomes of a microlevel variable may be mapped to outcomes of diffferent macrolevel variables, thus implying a splitting of the contribution of a microvariable across many macrovariables.

---

[1] This is formally a functor between $\mathcal{C}_{\mathcal{M}^m} \to \mathcal{C}_{\mathcal{M}^M}$ that maps objects and morphisms (or hom-sets), while preserving identity and composition.

[2] Invertibility on edges is always with reference to edges between nodes that are mapped under $\mathcal{C}_{\mathcal{M}^m} \to \mathcal{C}_{\mathcal{M}^M}$.

**Properties on the distributional layer.** Let us move on to consider the map $\mathcal{M}^m[\mathbf{X}] \to \mathcal{M}^M[\mathbf{X}']$. Differently from the previous map between graphs, we are now dealing with a map between sets that takes the form of a *function*. As before, let us investigate the properties that can be enforced on this map:

- *Functionality:* this property requires all the outcomes of the micromodel $\mathcal{M}^m$ to be mapped onto outcomes of the macromodel $\mathcal{M}^M$; there is no microlevel outcome, no matter how unlikely, that can be dropped and ignored in the macrolevel outcomes. Beyond a conceptual justification, a mathematical reason underpins this setup: a (measurable) function is necessary to guarantee a proper pushforward of the probability distribution over the outcomes of the micromodel $\mathcal{M}^m$ onto the outcomes of the macromodel $\mathcal{M}^M$; Functionality is a common assumption shared so far by all works on abstraction.

  Dropping this assumption would possibly require us to define a set of relevant outcomes to be mapped (in analogy with the definition of a set of relevant variable $R$ as a domain for the structural-layer mapping $a$ between variables in Rischel [2020]), together with a renormalization before or after the pushforward. This form of abstraction may reflect a mapping in which we capture the behaviour of a micromodel over a specific domain of outcomes, ignoring perhaps limit cases.

- *(Functional) surjectivity:* surjectivity requires that every outcome of the macromodel $\mathcal{M}^M$ is mapped from the domain of outcomes of the micromodel $\mathcal{M}^m$. This expresses the understanding that all possible macrolevel outcomes are explained at the microlevel. This property is usually considered a staple of abstraction, and for this reason it is introduced by Beckers and Halpern [2019] in their definition of $\tau$-abstraction, and it is enforced from the beginning in Rischel [2020]. Dropping surjectivity allows for some macrolevel outcomes without an explanation in the micromodel. This is accepted in Rubenstein et al. [2017] and, implicitly, in Otsuka and Saigo [2022].

- *(Functional) injectivity:* injectivity requires all the outcomes of the micromodel $\mathcal{M}^m$ to be mapped to different outcomes in the macromodel $\mathcal{M}^M$. This leads to a form of abstraction where the outcomes at microlevel are embedded into the set of outcomes at macrolevel. No work in the literature enforces this property.

- *(Functional) bijectivity:* a trivial property of bijectivity follows from surjectivity and injectivity. This would allow for abstractions where there is an isomorphism (an identity or a permutation) between micromodel and macromodel outcomes. Every outcome in the micromodel would have a unique corresponding outcome in the macromodel with the exact same probability. No

work in the literature enforces this property.

A property of invertibility would follow from bijectivity and surjectivity; furthermore, as before, two additional properties, normally taken for granted, are:

- *Micro-to-macro:* instead of assuming the directionality of an abstraction being $\mathcal{M}^m[\mathbf{X}] \to \mathcal{M}^M[\mathbf{X}']$, we could consider possibly stochastic maps $\mathcal{M}^M[\mathbf{X}'] \to \mathcal{M}^m[\mathbf{X}]$ going from the outcomes of the macromodels to the (sets of) outcomes of the micromodel.

- *Determinism:* the assumption of determinism reflects the understanding that the mapping from outcomes in the micromodel to outcomes in the macromodel expresses an exact deterministic relationship; if we were to drop this assumption, such as in the case in which we were to have limited knowledge, we could express the map $\mathcal{M}^m[\mathbf{X}] \to \mathcal{M}^M[\mathbf{X}']$ as a stochastic map. This would allow for forms of abstraction in which the contribution of an outcome in the micromodel would be split among a collection of outcomes in the macromodel; this could capture, for instance, our uncertainty on how an outcome at the microlevel would correspond to an outcome at the macrolevel.

## 4 DISCUSSION

Abstractions, like SCMs, are complex mathematical objects encoding not just statistical information but also causal assumptions normally expressed in the form of a DAG. Relying on a definition of an abstraction on the structural and distributional layer (as proposed by Otsuka and Saigo [2022], Rischel [2020], Rischel and Weichwald [2021]) guarantees a more fine-grained control on the definition of the form of an abstraction compared to approaches that focus only on the distributional layer (as done by Rubenstein et al. [2017], Beckers and Halpern [2019], Beckers et al. [2020]).

Although some definitions of abstraction do not explicitly characterize the abstraction on the structural layer, information about the structure of the graphs underlying the SCMs is not generally discarded but mediated by consistency. In Rubenstein et al. [2017], although knowledge of the structure of the SCM is not necessary, the distributions over the intervened models generated by the set $\mathcal{I}$ of interventions of interest force the structure of the SCM $\mathcal{M}^m$ in the $\mathcal{I}$-Markov equivalence class [Hauser and Bühlmann, 2015]; moreover, correspondence between distributions as required by consistency impose a further constraint on the form of the micrograph and the macrograph. However, the structural relationship is left implicit, and no formal properties may be requested for a mapping on the structural layer. The progressive strengthening in Beckers and Halpern [2019] of the original definition of $(\tau\text{-}\omega)$-abstraction may be seen as a way to sufficiently constrain the distributional-layer

definition so that undesired mappings between incompatible models whose distributions had been artificially set to satisfy the requirement of consistency are ruled out.

By explicitly considering the structural and the distributional layer, we have seen how enforcing different properties on the two layers of an abstraction may allow us to enlarge or restrict the family of transformations that can be considered as legitimate abstractions. Table 1 and 2 in Appendix A.1 summarize the properties that we defined and the types of abstractions that would be admitted by adding or dropping specific requirements. For instance, requesting the property of functoriality on the structural layer rules out the possibility that an abstraction could simplify a model by dropping causal links with limited strength. If such a simplification does not fit our understanding of abstraction, the property of functoriality could be part of the definition of abstraction. Appendix A.2 provides a list of illustrative examples of abstractions. The present treatment has highlighted which types of transformations would be preserved or rejected through the enforcement of specific properties, but it does not argue which ones should be selected.

Furthermore, the distinction between the structural and distributional layers also brings clarity on how an abstraction could independently act on the two layers to produce different results. *Coarsening* is a term often used as a generic synonym for abstraction. Yet, a coarsening may act on the structural or distributional layer. A coarsening on the structural layer implies a reduction in the number of observables. A coarsening on the distributional layer implies a reduction in the resolution of the observables. Similarly, a coarsening on the structural layer may be paired with an identity on the distributional layer; this would mean that the number of observables is reduced, but the resolution or the dimensionality of the new macrolevel observables is large enough that we can have a one-to-one map of microlevel outcomes to macrolevel outcomes.

Interestingly, the discussion on abstraction has focused on graphs and sets, and never explicitly on distributions and mechanisms. Distributions are complex objects to map, and focusing on the underlying set simplifies the problem; a map between microlevel and macrolevel sets entails an automatic pushforward of the microlevel distributions. Mechanisms are instead completely underdetermined; in its definition, an abstraction does not express any constraint on the macrolevel mechanisms; possible constraints are instead introduced via the requirement of consistency.

Indeed, a complete operative specification of abstraction can not preclude a definition of the consistency properties that an abstraction is supposed to guarantee. However, this paper has highlighted that even restricting our attention to the formal properties of an abstraction, there is a wide degree of freedom in defining what class of transformations should be considered an abstraction. This understanding has practical consequences: selecting the right properties would be relevant in any learning effort, as this may reduce the space of functions (or functors) over which we want to search for an abstraction.

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

# A APPENDIX

## A.1 SUMMARY OF PROPERTIES

Table 1 and 2 at the end of this appendix provide a quick visual summary of the properties we discusses and the types of abstractions they allow.

## A.2 EXAMPLES OF ABSTRACTIONS

This section provides examples of abstractions satisfying or contravening the properties discussed in the paper. We will use as a reference simple SCMs representing toy models for a lung cancer scenario. All examples are purely illustrative, and do not claim any scientific validity.

### A.2.1 Structural properties with respect to the nodes

We first present examples concerned with structural properties among nodes. Both the base model $\mathcal{M}^m$ and the abstracted model $\mathcal{M}^M$ will be defined on sets of variables including *smoking habit* $(S, S')$, *tar deposits* $(T, T')$, *lung cancer* $(C, C')$, *air pollution* $(P, P')$, *environmental factors* $(E, E')$.

**Functionality.** Figure 1 shows examples of abstraction concerned with the property of functionality among the nodes. The abstraction in Figure 1a is defined by the following mappings between the nodes: $S \mapsto S', T \mapsto S', C \mapsto C'$. This abstraction satisfies functionality as every node in the base model is mapped to a node in the abstracted model. This abstraction is also surjective and non-injective. The abstraction in Figure 1b, is defined, instead, by the abstraction mapping between the nodes: $S \mapsto S', C \mapsto C'$. As such, it is not a function, since the mediating node $T$ is ignored and not explicitly mapped onto any high-level variable. Restricting our abstraction map to the set of mapped nodes $(S, C)$, this abstraction is surjective and injective.

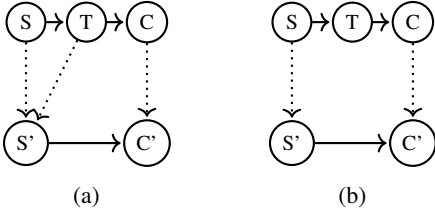

(a)  (b)

Figure 1: Functionality

Functionality expresses whether all the variables contribute to the definition of the high-level model (as in Figure 1a), or whether some of them may simply be ignored (as in Figure 1b).

**Functional surjectivity.** Figure 2 shows examples of abstraction concerned with the property of functional surjectivity among the nodes. The abstraction in Figure 2a is defined by the following mappings between nodes: $S \mapsto S', P \mapsto S', C \mapsto C'$. This abstraction satisfies surjectivity as every node in the abstracted model $(S', C')$ is determined by one or more nodes in the base model. However, with respect to all the nodes in the base model, this abstraction is non-functional; with respect to the set of mapped nodes $(P, S, C)$, instead, it is also non-injective. The abstraction in Figure 2b is defined, instead, by the abstraction mapping between nodes: $S \mapsto S', C \mapsto C'$. As such, it does not satisfies surjectivity, since a confounding node in the abstracted model $(E')$ is not explicitly mapped by any low-level variable. Again, this abstraction is, in general, non-functional; with respect to the set of mapped nodes $(S, C)$, however, it is injective.

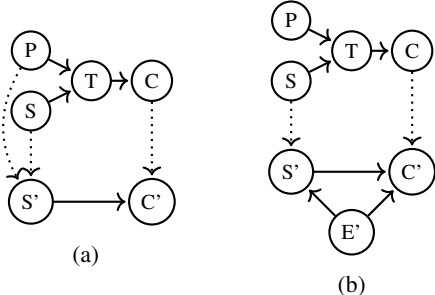

(a)

(b)

Figure 2: Surjectivity

Functional surjectivity expresses whether all the high-level variables are determined by low-level variables (as in Figure 2a), or whether some of them may have no corresponding variable in the low-level model (as in Figure 2b).

**Functional injectivity.** Figure 3 shows examples of abstraction concerned with the property of functional injectivity among the nodes. The abstraction in Figure 3a is defined by the following mappings between the nodes: $S \mapsto T', T \mapsto S', C \mapsto C'$. This abstraction satisfies injectivity as every node in the base model $(S, T, C)$ is mapped onto a different node in the abstracted model $(T', S', C')$. This abstraction is also functional and surjective. The abstraction in Figure 3b is defined, instead, by the abstraction mapping between nodes: $S \mapsto S', T \mapsto S', C \mapsto C'$. As such, it does not satisfies injectivity since two low-level variables $(S, T)$ are mapped on the same high-level variable $(S')$. This abstraction is also functional and surjective.

Functional injectivity expresses whether all the low-level variables are mapped to a distinct high-level variables (as in Figure 3a), or whether collapsing of variables is allowed (as in Figure 3b).

**Functional bijectivity.** Figure 4 shows examples of abstraction concerned with the property of functional bijec-

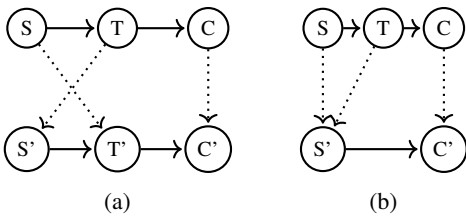

(a)            (b)

Figure 3: Injectivity

tivity among the nodes. The abstraction in Figure 4a is defined by the following mappings between the nodes: $S \mapsto S', T \mapsto T', C \mapsto C'$. This abstraction satisfies bijectivity as we have a one-to-one mapping between the low-level and high-level nodes. The abstraction in Figure 4b is defined, instead, by the abstraction mapping between nodes: $S \mapsto S', C \mapsto C'$. As such, it does not satisfies bijectivity because of a lack of a one-to-one correspondence.

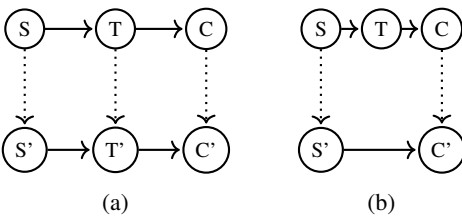

(a)            (b)

Figure 4: Bijectivity

Functional bijectivity expresses whether there should be a strict one-to-one correspondence between nodes in the low-level and high-level model (as in Figure 4a), or whether differences are allowed (as in Figure 4b). Notice that, in case of bijection among the nodes, an abstraction can lead to a simplification of the low-level model with respect to the outcomes of the macro-variables or the form of the mechanisms in the macromodel.

### A.2.2 Structural properties with respect to the edges

We now move to examine examples concerned with structural properties among edges. For simplicity, we will denote arrows in the base model $\mathcal{M}^m$ and the abstracted model $\mathcal{M}^M$ using an exponential notation: for instance, with reference to Figure 4a, we use $S^T$ to represent the edge from $S$ to $T$, $T^T$ to represent the identity edge from $T$ to $T$, and $S^{T^C}$ to represent the edge from $S$ to $C$ given by the composition of $S^T$ and $T^C$.

**Functoriality.** Figure 5 shows examples of abstraction concerned with the property of functoriality. In Figure 5a, let the abstraction mapping between the nodes be defined as in the case of Figure 4b. In the base model, the hom-set of relevant edges (relatively to the mapped nodes) is $\{S^S, S^{T^C}, C^C\}$; in the abstracted model the hom-set of

edges is $\{S'^{S'}, S'^{C'}, C'^{C'}\}$. Let the mapping between edges be defined as follows: $S^S \mapsto S'^{S'}, S^{T^S} \mapsto S'^{C'}, C^C \mapsto C'^{C'}$. This abstraction satisfies functoriality as every edge in the base model is mapped and composition is preserved. Furthermore, this abstraction is full and faithful. Let the abstraction in Figure 5b be defined, instead, by the mapping between nodes $S^S \mapsto S'^{S'}, S^{T^S} \mapsto C'^{S'}, C^C \mapsto C'^{C'}$. This abstraction is not functorial because composition is not preserved due to the different directionality of the arrows in the macromodel.

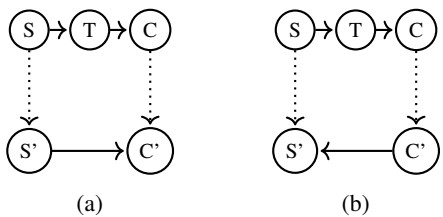

(a)            (b)

Figure 5: Functoriality

Functoriality expresses whether all the cause-effects links in the base model must be mapped and preserved in their directionality (as in Figure 5a), or whether some of them may be dropped or reversed (as in Figure 5b).

**Functorial fullness.** Figure 6 shows examples of abstraction concerned with the property of functorial fullness. In Figure 6a, let the abstraction mapping between the nodes be defined as in the case of Figure 4a. Let the abstraction between edges be defined as follows: $S^S \mapsto S'^{S'}, S^T \mapsto S'^{T'}, T^T \mapsto T'^{T'}, T^C \mapsto T'^{C'}, C^C \mapsto C'^{C'}, S^{T^C} \mapsto S'^{T'^{C'}}$. The abstraction is fully functorial because it is functorial, and the mapping between edges is surjective. Let the abstraction in Figure 6b be defined by the same mapping on nodes and edges. This abstraction is not fully functorial because the hom-set of the abstracted model contains an edge $S'^{C'}$ which is not mapped by any edge in the base model. Notice that this additional edge represent a direct effect $S'^{C'}$ which is different from the mediated effect $S'^{T'^{C'}}$.

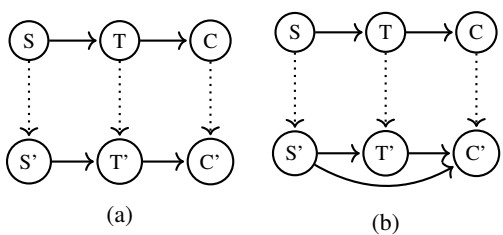

(a)            (b)

Figure 6: Functorial faithfullness

Full functoriality expresses whether the existence and direction of all high-level edges are determined by low-level edges (as in Figure 6a), or whether some of them may have no corresponding edge in the low-level model (as in Figure 6b).

**Functorial faithfullness.** Figure 7 shows examples of abstraction concerned with the property of functorial faithfullness. In Figure 7a, let the abstraction mapping between the nodes be defined as in the case of Figure 5a. In particular, let the abstraction between edges be defined as follows: $S^S \mapsto S'^{S'}, S^{T^S} \mapsto S'^{C'}, C^C \mapsto C'^{C'}$. The abstraction is faithfully functorial because it is functorial, and the mapping between edges is injective. Let the abstraction in Figure 7b be defined by the following mapping between edges: $S^S \mapsto S'^{S'}, S^T \mapsto S'^{S'}, T^T \mapsto S'^{S'}, T^C \mapsto S'^{C'}, C^C \mapsto C'^{C'}, S^{T^S} \mapsto S'^{C'}$. This abstraction is not faithfully functorial because the hom-set of the base model contains multiple edges $(S^S, T^T, S^T)$ mapped onto the same abstracted edge $(S'^{S'})$.

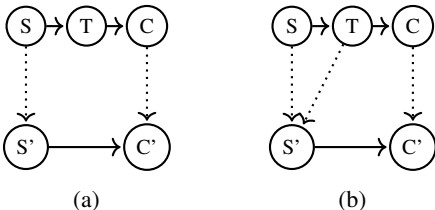

(a)                    (b)

Figure 7: Functorial fullness

Faithful functoriality expresses whether all low-level edges are mapped to distinct high-level edges (as in Figure 7a), or whether collapsing of edges is allowed (as in Figure 7b).

**Functorial full faithfullness.** Figure 8 shows examples of abstraction concerned with the property of functorial full faithfullness. In Figure 8a, let the abstraction mapping between the nodes be defined as in the case of Figure 6a. The abstraction is fully faithful because it is functorial, and the mapping between edges is bijective. Let the abstraction in Figure 8b be defined as in the case of Figure 7b. This abstraction is not fully faithful because there is not bijection between the hom-set of edges in the low-level model and the high-level model.

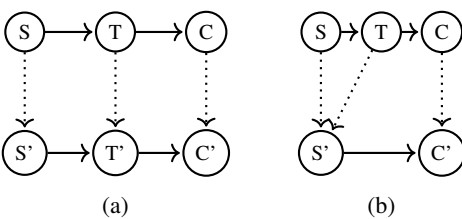

(a)                    (b)

Figure 8: Functorial full faithfullness

Fully faithful functoriality expresses whether there should be a one-to-one mapping between edges (as in Figure 8a), or whether a non-bijective map is allowed (as in Figure 8b). Similarly to the case of bijection among nodes, fully faithful functoriality narrowly constrains the form of the abstracted model; simplification of the base model is still possible, though, in terms of a reduction of the outcome

space of the macro-variables or the form of mechanisms in the macromodel.

### A.2.3  Distributional properties

Let us finally consider mappings on the distributional layer. We will here consider a single mapping between the domain $\mathcal{M}[S]$ of the smoking variable in the base model and the domain $\mathcal{M}'[S']$ of the smoking variable in the abstracted model; we will take into consideration both binary domains (simply representing whether a subject is a smoker or not) or multi-value domains (encoding the amount of smoking of a subject on a pre-defined scale).

**Functionality.** Figure 9 shows examples of abstraction concerned with the property of functionality among the outcomes. The abstraction in Figure 9a is defined by the following mappings between the outcomes: $0 \mapsto 0, 1 \mapsto 0, 2 \mapsto 1$. This abstraction satisfies functionality as every outcome in the base model is mapped to an outcome in the abstracted model. Furthermore, this abstraction is also surjective and non-injective. The abstraction in Figure 9b instead is not a function, since the outcome 0 in the base model is not mapped onto any high-level outcome. Restricting our abstraction map to the set of mapped outcomes $(1, 2)$, this abstraction is surjective and injective.

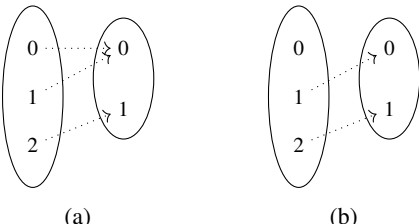

(a)                    (b)

Figure 9: Functionality

Functionality expresses whether all the outcomes are reflected in the high-level model (as in Figure 9a), or whether some of them may be dropped (as in Figure 9b).

**Surjectivity.** Figure 10 shows examples of abstraction concerned with the property of surjectivity among the outcomes. The abstraction in Figure 10a satisfies functionality as every outcome in the abstracted model is mapped from one or more outcomes in the base model. This abstraction is also non-injective. The abstraction in Figure 10b instead is not surjective, since the outcome 2 in the abstracted model is not mapped by any low-level outcome. This abstraction is also non-injective.

Surjectivity expresses expresses whether all the high-level outcomes are determined by low-level outcomes (as in Figure 10a), or some of them may have no corresponding outcome in the low-level model (as in Figure 10b).

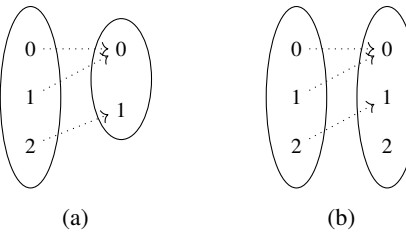

(a)          (b)

Figure 10: Surjectivity

high-level model (as in Figure 12a), or whether differences are allowed (as in Figure 12b). Notice that, in case of bijection among the outcomes, an abstraction still allows room for simplification in the form of the mechanisms of the macromodel.

**Injectivity.** Figure 11 shows examples of abstraction concerned with the property of injectivity among the outcomes. The abstraction in Figure 11a satisfies injectivity as every outcome in the base model is mapped to a distinct outcome in the abstracted model. This abstraction is also non-surjective. The abstraction in Figure 11b instead is not injective, since two outcomes $0, 1$ in the base model are mapped onto the same low-level outcome $0$. This abstraction is also non-surjective.

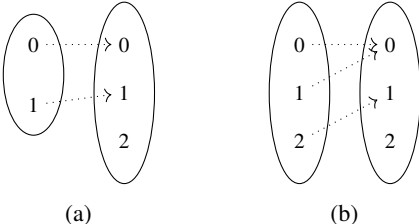

(a)          (b)

Figure 11: Injectivity

Injectivity expresses whether all the low-level outcomes are mapped to a distinct high-level outcome (as in Figure 11a), or whether collapsing of outcomes is allowed (as in Figure 11b).

**Bijectivity.** Figure 12 shows examples of abstraction concerned with the property of bijectivity among the outcomes. The abstraction in Figure 12a satisfies bijectivity because of the one-to-one mapping of outcomes in the base model and the abstracted model. The abstraction in Figure 12b instead is not bijective, because of the lack of injectivity among the outcomes.

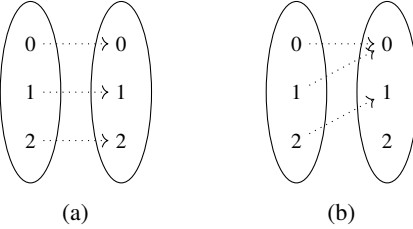

(a)          (b)

Figure 12: Bijectivity

Bijectivity expresses whether there should be a strict one-to-one correspondence between nodes in the low-level and

| | Identity | Node permutation | Node coarsening | Edge coarsening | Node embedding | Edge embedding | Node dropping | Edge dropping | Causal reversal | Causal splitting | Abs. Reversal |
|---|---|---|---|---|---|---|---|---|---|---|---|
| Functionality | ✓ | ✓ | ✓ | ✓ | ✓ | ✓ | × | ✓ | ✓ | × | - |
| Surjectivity | ✓ | ✓ | ✓ | ✓ | × | ✓ | × | ✓ | ✓ | × | - |
| Injectivity | ✓ | ✓ | × | ✓ | ✓ | ✓ | × | ✓ | ✓ | × | - |
| Bijectivity | ✓ | ✓ | × | ✓ | × | ✓ | × | ✓ | ✓ | × | - |
| Functoriality | ✓ | × | ✓ | ✓ | ✓ | ✓ | × | × | × | × | - |
| Fullness | ✓ | × | ✓ | ✓ | ✓ | × | × | × | × | × | - |
| Faithfulness | ✓ | × | ✓ | × | ✓ | ✓ | × | × | × | × | - |
| Fully Faithfulness | ✓ | × | ✓ | × | ✓ | × | × | × | × | × | - |
| Non-Determinism | - | - | - | - | - | - | - | - | - | ✓ | - |
| Macro-to-micro | - | - | - | - | - | - | - | - | - | - | ✓ |

Table 1: Summary of structural-layer properties and types of abstraction. The table expresses whether enforcing a specific structural-layer property allows for the definition of certain types of abstraction.

In the first row, a color-coded example of each type of abstraction is shown; the graphs of two simple chain-like models are presented: a micromodel on the left, and the corresponding macromodel on the right. The color of the nodes (and the edges, where relevant) in the micromodel illustrate to which corresponding nodes (and edges) of the macromodel they are mapped onto.

Observe that an *identity abstraction* is allowed under the enforcement of any of the properties. A *node permutation abstraction* is, in general, not allowed when enforcing functoriality, as it breaks the compositionality of edges. A *node coarsening abstraction* is possible if we do not require injectivity, while a *edge coarsening abstraction* is allowed if we do not require faithfulness. A *node embedding abstraction* requires surjectivity, while an *edge embedding abstraction* necessitates no fullness. *Node dropping abstraction* is not allowed under any property. *Edge dropping abstraction* and *causal reversal abstraction* are allowed if we do not require functoriality. Notice that the bijectivity row and the fully faithfulness row are just derived from surjectivity and injectivity or from fullness and faithfulness. Finally, the last two rows and columns consider less common properties and scenarios. A *causal splitting abstraction*, depicted by having a macrolevel node colored purple because it is mapped by both the blue and red microlevel nodes, requires a non-deterministic map. An *abstraction reversal*, represented by a double arrow going from the macrolevel to the microlevel, requires a structural-layer map going from the graph of the macromodel to the graph of the micromodel.

| | Identity/Permutation | Coarsening | Embedding | Outcome dropping | Outcome splitting | Abstraction reversal |
|---|---|---|---|---|---|---|
| Functionality | ✓ | ✓ | ✓ | × | × | × |
| Surjectivity | ✓ | ✓ | × | × | × | × |
| Injectivity | ✓ | × | ✓ | × | × | × |
| Bijectivity | ✓ | × | × | × | × | × |
| Non-Determinism | - | - | - | - | ✓ | - |
| Macro-to-micro | - | - | - | - | - | ✓ |

Table 2: Summary of distributional-layer properties and types of abstraction. The table expresses whether enforcing a specific distributional-layer property allows for the definition of certain types of abstraction.

In the first row, a color-coded example of each type of abstraction is shown; two simple sets are presented: the set of micromodel outcomes on the left, and the set of macromodel outcomes on the right. The color of the elements (crosses) in the micromodel set encodes the mapping onto elements (crosses) of the macromodel with the corresponding color.

Observe that an *identity/permutation abstraction* is allowed under the enforcement of any of the properties. A *coarsening abstraction* is possible if we do not require injectivity. An *embedding abstraction* if we do not require surjectivity. An *outcome dropping abstraction* is not allowed under any property.

Notice that the bijectivity row is just derived from surjectivity and injectivity.

Finally the last two rows and columns consider less common properties and scenarios. A *outcome splitting abstraction*, depicted by having a macrolevel cross colored purple because it is mapped by both the blue and red microlevel elements, requires a non-deterministic map. An *abstraction reversal*, represented by a double arrow going from the macrolevel to the microlevel, requires a distributional-layer map going from the set of outcomes of the macromodel to the set of outcomes of the micromodel.