# OpenReview forum: "Abstraction between Structural Causal Models: A Review of Definitions and Properties"
_auai.org/UAI/2022/Workshop/CRL — CRL@UAI 2022 Oral_

### Official Review · Reviewer_tJjX · 2022-06-13

**Rating:** 7
**Confidence:** 4

**Review:**

### Summary of the paper
This paper proposes a classification scheme for abstractions of structural causal models based on 1) structural properties and 2) distributional properties. This framework allows to organize and compare different abstraction proposals from the literature as well as pointing to unexplored region of the space of possible abstractions.

### Disclaimer
I cannot really assess the novelty nor whether the literature review is sufficient since I am not well versed in this domain. Also, I could not follow some of the (few) references to category theory.

### Review
Overall, I enjoyed reading the paper and I think it will be of interest to the UAI community. I believe the framework proposed will influence my future thinking when it comes to causal abstraction. I believe the manuscript could be greatly improved by adding examples to illustrate the various categories of abstractions. I recommend acceptance.

First point of the second page: $X_i$ is defined over a _finite_ domain? It does not seem necessary throughout the paper.

Assumption (d) in 2.1: The definition of SCM above says that "$\mathcal{P}$ is a set of N probability distributions $p_i$, one for each exogenous variable", which suggests that the endogenous noises are independent, contradicting (d). If you want to allow for dependent noises, the definition of SCM should include a _joint distribution_ over noises, otherwise the SCM is not well defined (how does one go from the set of distributions $p_i$ to the joint over the noises?).

In the last sentence of the paragraph on Rischel [2020] in the literature review, is there a prime missing? I think it should be "$\alpha_{X'}: \mathcal{M}^m[a^{-1}(X_i')] \rightarrow ...$", although I am not certain.

**Properties at the structural level:**

Functional bijectivity: At first, I had a hard time finding an example that would satisfy this condition while still qualifying as an abstraction. Maybe provide an example? I thought of a case where we discretize variables while keeping the graph the same.

Functorial fullness: What are "hom-sets"?

**Properties at the distributional level:**

What would be an example on an interesting abstraction which is bijective at the distributional level?

---

### Official Review · Reviewer_fLyC · 2022-07-03
**Review of "Abstraction between Structural Causal Models: A Review of Definitions and Properties"**

**Rating:** 7
**Confidence:** 4

**Review:**

The authors provide a review of recent proposals for a notion of “abstraction” from a micro-level causal model to a macro-level causal model. They introduce a categorization scheme to compare these proposals, with three main properties under comparison. First, they categorize the relation between nodes in the micro-level model and the macro-level model. Second, they categorize the relation between edges in the micro and macro models, in terms of functoriality of the abstraction. Finally, they categorize the relations between the possible values of variables in the two models. They dissect the constraints implied by these properties, nicely summarized in a table in the supplementary material.

The paper serves a useful role, laying out a significant proportion of the design space for abstraction. It is also clearly written, and provides to the best of my knowledge a quite thorough review of recent work. By the nature of the topic, the paper is not introducing anything original; even the categorization of these properties has been touched on in previous work, especially in Beckers and Halpern (2019). However, there is, to the best of my knowledge, no similar review of causal abstraction, making the paper novel in that respect.

To conclude, we have the following:

**Pros:**
- The paper provides a thorough review of relevant literature.
- The paper is clearly organized and the writing is easy to follow.

**Cons:**
- The paper lacks any concrete examples, which limits its significance. Examples would make this paper a very solid review.

**Minor suggestions:**
- Create a table to categorize the existing notions of abstraction.
- Shorten the sections on injectivity/bijectivity (analogously, functiorial faithfulness/fully faithfulness), since these are not of much interest in a notion of abstraction.
- I was confused by two sentences. First, before Section 2.2, the authors say “we can conveniently represent the SCM as a collection of sets and a collection of mechanisms”. How are these related to the standard representation of the SCM? Second, in the second-to-last paragraph of Section 2.3, the authors define surjective maps for each macrovariable. It is unclear to me that the maps type-check: the authors take $a^{-1}$ of $X_i$, but it seems $X_i$ is  a micro-variable while $a^{-1}$ should take a macro-variable as an argument.

---

### Meta-Review · Program_Chairs · 2022-07-05

**Recommendation:** Accept (Oral)
**Confidence:** 4

**Metareview:**

The paper provides an interesting and very relevant contribution to the workshop, as both of the reviewers agree.
Both also provide very useful suggestions to make it even better and more accessible to the UAI community.

---

### Decision · Program_Chairs · 2022-07-06

Accept (Oral)